# Study on the Improved Method of Urban Subcatchments Division Based on Aspect and Slope- Taking SWMM Model as Example

**Zening Wu, Bingyan Ma** **, Huiliang Wang and Caihong Hu \***

School of Water Conservancy Engineering, Zhengzhou University, Zhengzhou 450001, China; zeningwu@zzu.edu.cn (Z.W.); mabingyan@gs.zzu.edu.cn (B.M.); wanghuiliang@zzu.edu.cn (H.W.)
**\*** Correspondence: hucaihong@zzu.edu.cn; Tel.: +86-187-3991-9805

**Abstract:** The storm water management model (SWMM) is widely used in urban rainfall runoff simulations, but there are no clear rules for the division of its sub catchment areas. At present, the popular sub catchment area division method takes the average slope as the slope parameter of the sub catchment area, which brings errors to the model in mechanism. Based on the current method, this paper proposes a new method to further subdivide the sub catchment area of the SWMM model, according to the Digital Elevation Model (DEM) data of underlying surface, slope and aspect information. By comparing with the previous methods, it was found that the division method based on slope and aspect can make the setting of model parameters and hydraulic exchange conditions clearer, and improve the accuracy of the model on a certain level.

**Keywords:** SWMM model; subcatchments division; dem data; slope & aspect; method improvement

## 1. Introduction

The storm water management model (SWMM) is a model proposed by the US Environmental Protection Agency (EPA) to solve urban rainstorm flooding. Since the development of the model in 1971, after more than 40 years application, the model has been recognized worldwide [1–3]. The subcatchment is the smallest hydrological response unit of the SWMM model, which directly affects the parameter settings and simulation results [4,5]. Reasonable subcatchment division is the most critical step in the construction of the SWMM model. Scholars have studied it [6,7], and roughly formed three schemes for dividing the subcatchment.

According to the trend of the pipe network, the distribution of streets and river channels, the direct manual division of subcatchments is the most popular division scheme at present [8–10]. This method uses actual urban roads, pipe networks and rivers as the boundary. The operation is relatively simple, and the physical meaning is clear. Roads, pipes and rivers are also the basis for the construction of different functional areas in the city. Therefore, by using this method, the social function of the catchment area is relatively single, but the underlying surface conditions are often different. The physical meaning of the underlying surface parameters during the modeling process is unclear and difficult to determine [11,12]. An improved method is to divide the urban subcatchment area according to the type of soil use [13]. This method makes the parameters have a clear physical meaning, which is easy to set, reflects the idea of distributed simulation and is more in line with the actual situation. But this method requires detailed underlying surface information, the operation is tedious and the problem of accumulation of errors caused by the increase in calculation volume cannot be avoided, so it is suitable for developed and smaller cities [14]. When the data is scarce or the research area is relatively large, the above two methods are no longer applicable, and the Tyson polygon method is used directly

to divide the subcatchment area, according to the distribution of the nodes [15]. This method does not consider land use, and the operation is simple, but the accuracy of the simulation is not as good as the above two methods [16]. For the convenience of discussion, this paper refers to the above three types as "Natural-artificial boundary method", "Land use method" and "Tyson polygon method".

As shown in Table 1, each of the three methods has its advantages and disadvantages. For most cities, the rainwater outlets are evenly distributed along the main road, and the rainwater pipe network connects all the rainwater outlets along the road direction. Therefore, a series of water collection units are divided by the pipe network-road network as the boundary. The rainwater of the water unit is discharged from the rainwater outlet on its boundary, and will not flow into the adjacent water storage unit. Therefore, the pipe network-road network-river network system has a blocking effect on the slope flow. The "Natural-artificial boundary method" determines that the subcatchment unit conforms to the physical laws of runoff, but the land use types and topographic changes in the subcatchment area cannot be considered, which brings difficulties to parameter setting and determination of water flow direction. The "Land Use Method" highlights the attributes of land use in the subcatchment area, that is, a relatively single land use type as a subcatchment area, which can be divided into four categories: grassland, forest land, water area and building land. The underlying surface parameters corresponding to the catchment area are easy to determine, and have a clear meaning, but the slope parameter is a special case, because the slope does not change with the type of soil use. Therefore, although this method solves most of the initial parameter setting problems, the problems of runoff path and runoff direction still cannot be solved, and the number and spatial complexity of its subcatchments are also the highest among the three methods. The calculation amount is large and calculation errors are easy to accumulate. In contrast, the "Tyson polygon method" ignores the spatial physical properties of the underlying surface, and its purpose is to place each rainwater outlet in the center of the subcatchment, which solves the problem of runoff paths to a certain extent. In the case of neglecting the changes in terrain and slope, this treatment makes the catchment range of the subcatchment area different from the actual situation [17,18].

**Table 1.** Main division methods of subcatchments and their advantages and disadvantages.

| Methods | Advantages | Disadvantages | Cases |
|---------|-----------|---------------|-------|
| Natural-artificial boundary method | Clear physical mechanism, moderate data volume, moderate modeling workload | Difficult to determine parameters and runoff paths | Surat, India [8] West Azerbaijan province, I.R.Iran [9] Kunming Dongfeng East Road catchment area, China [10] |
| Land use method | The physical mechanism is clear, and the underlying surface parameters are easy to determine | Large amount of data, difficult to determine runoff paths, and large modeling workload | Jinan, China [13] |
| Tyson polygon method | Less data requirements, less modeling effort, and relatively clear runoff paths | The physical mechanism is not clear, and it is difficult to determine the underlying surface parameters | Dikrong, India [15] |

In summary, the main problem of the current mainstream subcatchment division methods are that the topographic features of the subcatchment area are represented by the average slope, which cannot accurately reflect the impact of topographical conditions on the subcatchment, so that the simulated runoff deviate from the actual situation. As the key parameters for the calculation of runoff and sink, slope and aspect must be considered in the smallest hydrological response unit, in order to obtain subcatchments with relatively stable slope and aspect.

Based on the "Natural-artificial boundary method", this study proposes a method for subcatchment division of an improved SWMM model based on the Digital Elevation Model (DEM) data. The method takes into account the terrain gradient factors in the subcatchment area with roads, pipe networks and rivers as the boundary. According to the underlying DEM, the subcatchment area with large terrain fluctuations is further divided into multiple smaller slope properties. The small subcatchment area has improved the method of dividing the subcatchment area, according to the trend of the pipe network, the distribution of streets and river channels to a certain extent. The application on the new campus of Zhengzhou University shows that this method reduces the workload of parameter calibration and improves the accuracy of urban flood simulation.

Section 2 introduces the improved method and its rationality. Section 3 is the application of the method. The application results and comparison are shown in Section 4. Sections 5 and 6 analyze and discusses the methods and results.

## 2. Methods

The three subcatchment division methods of the SWMM model described in the introduction have a common feature. The slope of each subcatchment is generally expressed as the average slope of the subcatchment, which cannot reflect the continuous slope of the underlying changes, which bring some errors to the model's flow and velocity calculations.

### 2.1. Error Caused by Average Slope

SWMM solves the continuous equation (Equation (1)) and the Manning equation (Equation (2)) simultaneously, and calculates surface confluence according to the nonlinear reservoir method [19].

$$\frac{dV}{dt} = Ai^* - Q \tag{1}$$

$$Q = \frac{W}{n}\left(d - d_p\right)^{5/3} S^{1/2} \tag{2}$$

where, $V$ is the surface water collection volume, m³; $t$ is the time, s; $A$ is the surface area, m²; $i^*$ is the net rainfall intensity, mm/s; $Q$ is the outflow, m³/s; $W$ is the sub-basin flood width, m; $n$ is the surface Manning coefficient; $d$ is the water depth, m; $d_p$ is the maximum depth of surface water storage, m; $S$ is the average slope of the subcatchment. As shown in Figure 1, when $d > d_p$, there will be slope confluence, and the water within the depth of $d_p$ will not be discharged into the pipeline or river.

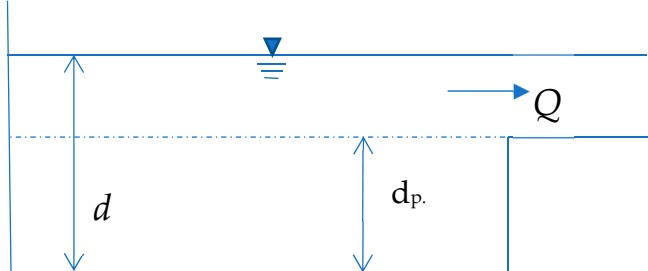

**Figure 1.** Slope confluence with water storage in depressions [19].

Equation (2) shows that the flow of the subcatchment is proportional to the power of 1/2 of the slope. For the slope parameter $S$ with a clear physical meaning, if the average value is used instead, the continuous variation characteristics of the slope in the subcatchment area will be ignored, and errors will occur due to parameter generalization. In theory, the topography of the subcatchment area will fluctuate. The larger the average slope, the greater the error.

The effect of slope and aspect on velocity is reflected in the micro terrain. As shown in Figures 2 and 3, it can be assumed that there is a subcatchment Z with road-pipes and network-rivers

as the boundary. The slope of the subcatchment zone is continuously changing. Now the underlying surface is continuously divided into $m$ rows and $n$ columns. Square grid, where the $i$-th row ($i \leq m$) and $j$ ($j \leq n$) column cell number is $Z_{ij}$, the adjacent cells are $Z_{i-1,j-1}$, $Z_{i-1,j}$, $Z_{i-1,j+1}$, $Z_{i,j-1}$, $Z_{i,j+1}$, $Z_{i+1,j-1}$, $Z_{i+1,j}$ and $Z_{i+1,j+1}$. Considering the boundary velocity of $Z_{i,j}$ and $Z_{i,j+1}$, the Equation (3) is as follows [20]:

$$u = \sqrt{g\left(\frac{l_1 - l_2}{r} - \frac{h_2 - h_1}{r}\right) \bullet \frac{\max(h_1, h_2)}{C_D}} \tag{3}$$

where, $u$ is the water flow velocity (m/s) at the boundary of two adjacent cells Zi, j and Zi, j + 1 with elevations $l_1$ and $l_2$ and side length $r$, $g$ is the acceleration of gravity (9.8 m/s$^2$), $C_D$ is the drag coefficient, $h_1$ and $h_2$ are the water depths of the two cells along the water flow direction. When the cell is small enough, according to the continuity characteristics of the water flow, it can be considered that the adjacent two cells have the same water depth, that is, $h_1$ and $h_2$ are equal to h, as shown in the Figure 3, and the slope $S$ is given by Equation (4):

$$S = \frac{l_1 - l_2}{r} \tag{4}$$

when $r$ is small enough, the water depths $h_1$ and $h_2$ of adjacent units are almost the same, considering the vector characteristics of slope and velocity, Equation (3) can be simplified into Equation (5):

$$\vec{u} = \sqrt{\frac{g\vec{S}h}{C_D}} \tag{5}$$

| | 1 | 2 | 3 | ... | j-1 | j | j+1 | ... | n |
|---|---|---|---|---|---|---|---|---|---|
| 1 | $Z_{1,1}$ | $Z_{1,2}$ | $Z_{1,3}$ | ... | $Z_{1,j-1}$ | $Z_{1,j}$ | $Z_{1,j+1}$ | ... | $Z_{1,n}$ |
| 2 | $Z_{2,1}$ | $Z_{2,2}$ | $Z_{2,3}$ | ... | $Z_{2,j-1}$ | $Z_{2,j}$ | $Z_{2,j+1}$ | ... | $Z_{2,n}$ |
| 3 | $Z_{3,1}$ | $Z_{3,2}$ | $Z_{3,3}$ | ... | $Z_{3,j-1}$ | $Z_{3,j}$ | $Z_{3,j+1}$ | ... | $Z_{3,n}$ |
| ⋮ | ⋮ | ⋮ | ⋮ | | ⋮ | ⋮ | ⋮ | | ⋮ |
| i-1 | $Z_{i-1,1}$ | $Z_{i-1,2}$ | $Z_{i-1,3}$ | ... | $Z_{i-1,j-1}$ | $Z_{i-1,j}$ | $Z_{i-1,j+1}$ | ... | $Z_{i-1,n}$ |
| i | $Z_{i,1}$ | $Z_{i,2}$ | $Z_{i,3}$ | ... | $Z_{i,j-1}$ | $Z_{i,j}$ | $Z_{i,j+1}$ | ... | $Z_{i,n}$ |
| i+1 | $Z_{i+1,1}$ | $Z_{i+1,2}$ | $Z_{i+1,3}$ | ... | $Z_{i+1,j-1}$ | $Z_{i+1,j}$ | $Z_{i+1,j+1}$ | ... | $Z_{i+1,n}$ |
| ⋮ | ⋮ | ⋮ | ⋮ | | ⋮ | ⋮ | ⋮ | | ⋮ |
| m | $Z_{m,1}$ | $Z_{m,2}$ | $Zm,3$ | ... | $Z_{m,j-1}$ | $Z_{m,j}$ | $Z_{m,j+1}$ | ... | $Z_{m,n}$ |

**Figure 2.** Schematic diagram of cells in the subcatchment Z.

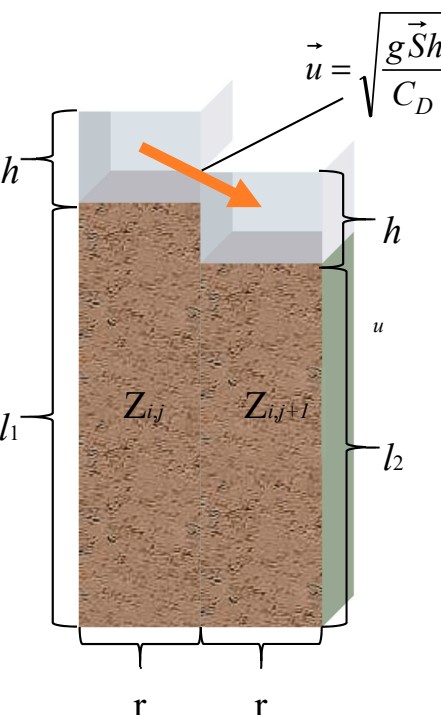

**Figure 3.** Schematic diagram of velocity and direction of adjacent cells.

Equation (5) indicates that the velocity at each location in the subcatchment area is also proportional to the 1/2 power of the slope, the slope and aspect determine the runoff and direction. In the subcatchment area with "pipe-road-river network" as the boundary, the topographical fluctuations change the flow velocity and direction of the slope water flow at the microscopic level. In the end, the subcatchment Z will hydraulically exchange to the surrounding subcatchments or nodes under the combined effect of all microscopic water flows. If the average slope is used as the slope parameter of the subcatchment, the microscopic physical process will be replaced by the macro concept. Not only does it cause errors in velocity calculation, but it also directly generalizes the direction of the water flow, which makes it difficult to determine the subcatchment interval and the hydraulic exchange relationship between the subcatchment area and the nodes, and also generates random errors in the amount of water exchange. In summary, reducing the error caused by the generalization of the average slope is of great significance to the calculation of the model.

*2.2. Improvement of Methods*

Roads, ditches and culverts influence hydrological and geomorphological processes significantly, DEM data with sufficient accuracy (the width of the highest level road in the area) is the key to flood simulation [21]. In this study, Global Positioning System (GPS), total station, and theodolite were used to acquire DEM data with a resolution of 2 m in the study area.

In the study of watershed scale, the Arcgis platform can be used to automatically identify sub-watersheds, based on DEM data, with sufficient accuracy. However, this method is not suitable for urban-scale flood simulation [22]. The conditions of the urban underlying surface are complex. Some small-scale micro-topography often has a great impact on the runoff process, such as rainwater wells and curbstones. The identification of micro-topography requires extremely high-precision DEM data (less than 0.5 m), which makes data acquisition and processing difficult. In addition, infrastructure such as pipe networks and roads undertake the main drainage tasks of the city, and subcatchments based on DEM identification in urban areas often fail to fully extract roads and pipe network, making the divided river channels and natural runoff channels exhibit a huge difference. Therefore, whether the

subcatchment is divided by the "Natural-artificial boundary method" or "DEM identification sub-basin method", the subcatchment of the real SWMM model cannot be obtained.

Under the premise that ultra-high-precision DEM data cannot be obtained, the combination of the "Natural-artificial boundary method" method and the "DEM identification sub-basin method" can effectively solve the above problems. Figure 4 shows the flow chart, considering the principle of "Water flowing to a low place" in the area with road-pipe network-rivers as the boundary, a reasonable subcatchment can be obtained. This study is based on the subcatchments divided by the "Natural-artificial boundary method" method, and these subcatchments are called "original subcatchments", and the representativeness of the elevations of these original subcatchments is analyzed. Due to the change of the slope direction in the subcatchment area, when the standard deviation of the elevation of the subcatchment area is not greater than 0.6, the terrain is considered to be gentle, and the average slope can be used as the slope parameter. Otherwise, the terrain changes significantly, and it is necessary to further divide the original subcatchment according to the method of DEM automatic subcatchment identification, so as to obtain new subcatchments with more representative slope parameters. It should be noted that the subcatchment elevation standard deviation threshold $\sigma$ is a value defined according to modeling needs. The smaller the value, the finer the subcatchments division, and the higher the accuracy of the simulation, but model building will be more difficult.

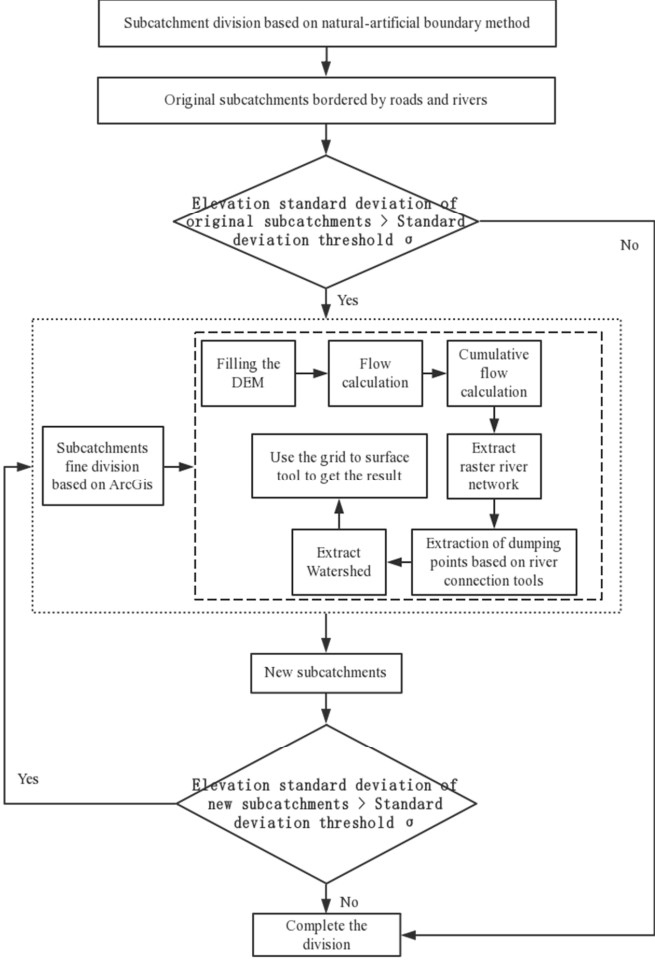

**Figure 4.** Flowchart of the methods.

The slope and aspect map of the study area can clarify the slope and aspect characteristics of the subcatchment. Further refinement of the original subcatchment area reduces the error caused by the average slope, and the addition of the aspect information reduces the structural error caused by

the unknown definition of the hydraulic exchange conditions in the subcatchment area. In general, the model is brought closer to the actual situation, and the model errors and uncertainties are controlled accordingly.

## 3. Method Application

### 3.1. Study Area

This study uses the new campus of Zhengzhou University (ZZU), China as the research area. The new campus of Zhengzhou University is located in the northwest of Zhengzhou City, Henan Province. Construction began on 28 August 2001, and all major projects were completed in 2005. The campus covers an area of about 2.3 km$^2$, about 2.1 km from north to south, and about 1.1 km from east to west, and is rectangular in shape. The east-west boundary is Changchun Road and the West Fourth Ring Road, the north-south boundary is Science Avenue and Lianhua Street, with a total construction area of 1.4 km$^2$. The rapid construction of the campus has caused drastic changes in the conditions of the underlying surface. Before the start of construction, the terrain was relatively flat, mainly farmland, grassland and woodland. After the start of construction, the conditions of the underlying surface changed dramatically, from a single green space to a comprehensive area integrating green land, slope land, water area and building land. In particular, the newly excavated artificial lake "Meihu" (about 800 m in length and 100 m in width and 0.08 km$^2$ in catchment area), and "Thick mountain" piled in earth with construction excavation (range about 0.0625 km$^2$, with an average height of about 20 m above the ground) and the construction land, which occupies 2/3 of the school's area, has greatly changed the natural runoff field.

At present, the drainage of rainwater in this area mainly depends on the rainwater pipe network, all the pipe networks are laid along the main road and finally enter the urban main pipe through the drainage outlet located at the northwest corner of the campus. The study area belongs to a temperate monsoon climate with an average annual rainfall of 542.15 mm. The rainfall in June–September accounts for more than 60% of the annual rainfall. Therefore, the risk of urban waterlogging in the region increases sharply every summer. In addition, with the expansion of the city, the area shows a clear urban rain island effect, and its rainfall is increasing, and the rainwater pipeline network has gradually failed to meet the requirements for timely removal of rainwater, which has led to frequent campus flooding, as shown in Figure 5. By collecting drainage network data, and analyzing the flood simulation results before and after the improvement of the subcatchment division method, the rationality of the method improvement was tested.

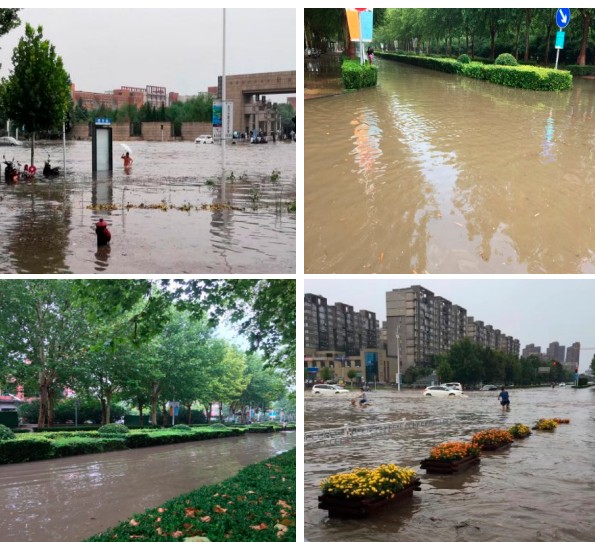

**Figure 5.** Waterlogging on the campus of Zhengzhou University in recent years.

*3.2. Data Acquisition and Processing*

3.2.1. Land Use and Rainfall Data

Land use and runoff data were obtained from school management. As shown in Figure 6, the green part is the campus greening land, mainly artificial turf, bushes and woodland, the yellow-green part is undeveloped bare land and wasteland, and the white part is construction land, mainly including roads and buildings. The blue area is the waters, including Meihu Lake in the center of the campus and the waterfall in the thick mountain in the north. The campus has a total of 371 rainwater outlets, and 37 main rainwater pipes are buried under the road for hydraulic exchange with the road through the rainwater outlets. All the pipe network water flows finally into the trunk pipe in the northwest corner of the campus and exits the campus. The rainfall data used in the study are measured data from the nearest rainfall station, number 50606605.

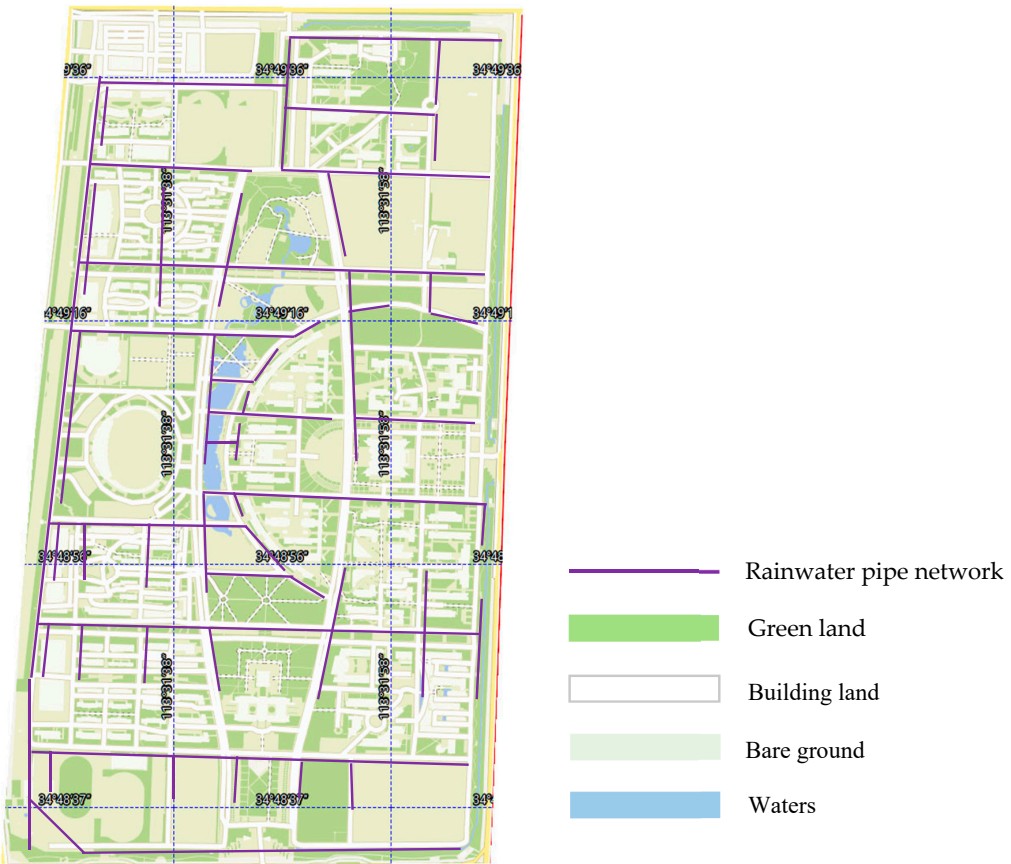

**Figure 6.** Land use of the study area.

3.2.2. DEM Data

The accuracy of DEM data directly affects the rationality of subcatchment division [8]. In this study, Global Positioning System (GPS), total station and theodolite were used to acquire DEM data, with a resolution of 2 m in the study area using ground survey methods.

The generated point elevation data set contains a large number of peaks, which is due to the presence of building structures, trees and other high-rise objects commonly found in urban environments. Therefore, the elevation dataset was edited to select elevation points in open areas (such as playgrounds, open spaces, fallow land, and wide roads) by using detailed land use/land cover datasets, while masking irregularities introduced in the DEM Area, thereby eliminating peaks. These points are further interpolated to generate a bare ground surface. By processing the DEM,

the bare surface is refined to make the DEM free of depressions. Because the terrain of the study area is a gentle slope, the steep slopes were identified and filtered out by smoothing.

### 3.3. Parameter Value

The parameters of the model were determined according to the SWMM model manual and the related literatures [23], as shown in Tables 2–4.

**Table 2.** Depression depth in different land use types.

| Land Use Types | Depression Depth Experience Value | Depression Depth Rated Value |
|---|---|---|
| Impervious surface | 1.27~2.54 | 2.10 |
| Grassland | 2.54~5.08 | 3.90 |
| Woodland | 7.62 | 7.62 |

**Table 3.** Surface Manning coefficient.

| Surface Type | Manning Coefficient | Surface Type | Manning Coefficient |
|---|---|---|---|
| Flat asphalt | 0.011 | Natural grassland | 0.13 |
| Flat concrete | 0.012 | Grass and forest | 0.14 |
| Cement gravel surface | 0.024 | Sparse grass | 0.15 |
| Building area <20% | 0.06 | Lush grassland | 0.24 |
| Building area> 20% | 0.17 | Sparse undergrowth | 0.4 |

**Table 4.** Manning coefficient of pressure pipelines.

| Pipeline Material | Manning Coefficient Empirical Value | Manning Coefficient Fixed Value |
|---|---|---|
| Asbestos cement pipe | 0.011~0.015 | 0.013 |
| Smooth cast iron pipe | 0.012~0.014 | 0.013 |
| Concrete pipe | 0.011~0.015 | 0.014 |
| Smooth plastic tube | 0.011~0.015 | 0.013 |

## 4. Results

### 4.1. Model Construction

The study area is divided into 40 subcatchments according to the "Natural-artificial boundary method", numbered Z1, Z2, Z3, . . . , Z40. As shown on the left of Figure 7, the processed DEM data is imported into ArcGis. The hydrological analysis tool was used to directly divide the sub-basin based on the principle of "Water flows to lower places". The results are shown in the right of Figure 7. The "Natural-artificial boundary method" is based on the road and water system as the boundary, which can reflect the distribution of the road-water system of the underlying surface. The second method is to identify the watershed based on DEM data and divide the underlying surface into subwatersheds. The comparison found that the "Natural-artificial boundary method" could not reflect the true flow path inside the subcatchment, and due to the limitation of DEM data accuracy, the DEM identified subcatchments could not reflect the blocking effect of roads and waters on runoff [22], nor could it reflect the micro-topography. The impact of runoff caused the two division results to be inconsistent with the actual flow process.

Taking the road and the water system as the boundary of the subcatchments are often easy to ignore the topographical features of the subcatchments. If the terrain fluctuates greatly, using the average slope to calculate runoff will cause a large error. Further division the subcatchments with large terrain fluctuations according to slope and aspect can effectively reduce this error. Since there is no large area with a uniform slope in the study area, the elevation standard deviation $\sigma$ can be used as an effective indicator to measure the topography of the subcatchments. The larger the $\sigma$, the greater the elevation change of the subcatchments, and the greater the terrain relief. On the contrary,

the smaller the σ, the smaller the terrain relief. Equation (6) is the elevation standard deviation formula of subcatchment Zm.

$$\sigma = \sqrt{\frac{1}{n-1}\sum_{i=1}^{n}\left(h_i - \bar{h}\right)^2} \tag{6}$$

where, σ is the standard deviation of the elevation of Zm, $n$ is the number of elevation measurement points in Zm, $h_i$ is the elevation of the $i$-th measurement point, and $\bar{h}$ is the average value of elevation of Zm.

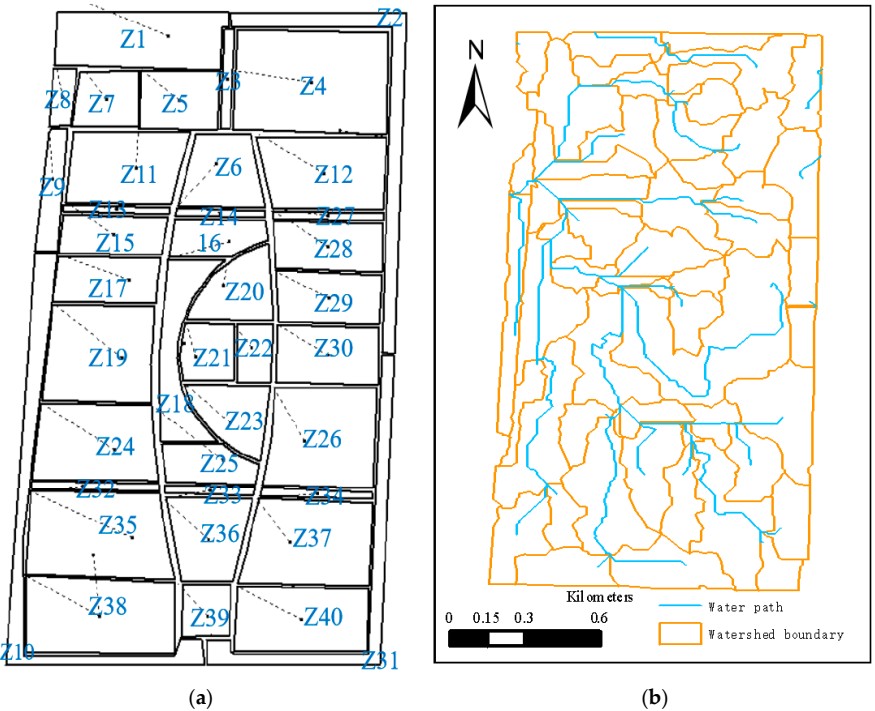

(**a**)　　　　　　　　　　　　　　　　　　　　(**b**)

**Figure 7.** Subcatchment area divided according to "road-pipe network" and sub-watershed area divided according to the Digital Elevation Model (DEM). (**a**) Subcatchments bounded by road-pipe network and waters; (**b**) Subcatchments based on DEM.

As shown in Figure 8, the 40 original subcatchments numbered Z1, Z2, … , Z40 in Figure 7a are divided into eight groups according to the numbering order, and five in each group. Plot the elevation data of 2 m resolution in the study area into a scatter plot. The 2 m resolution elevation data of the underlying surface is drawn into eight scatter plots. The horizontal coordinate of each figure is the number of elevation sampling points, and the vertical coordinate is the elevation. Different subcatchments in each group are used different colors indicate that the legend gives the standard deviation of elevation for each subcatchment. Due to the different size of each subcatchment, the number of elevation sampling points in each subcatchment area is also different. The larger the subcatchment, the more sampling points, and the more scattered the points are, the greater the relief is and the greater the corresponding σ is. In Figure 9, among the 40 original subcatchments, the standard elevation differences of Z2, Z4, Z6, Z10, Z12, and Z18 are greater than 0.6, and are significantly higher than other sub-catchments, indicating that the underlying surface is undulating, the average slope cannot be used as a slope parameter and needs to be further subdivided according to the actual situation. Here, 0.6 is used as the threshold of σ. If more subcatchments need to be divided, a smaller number can be used as the threshold.

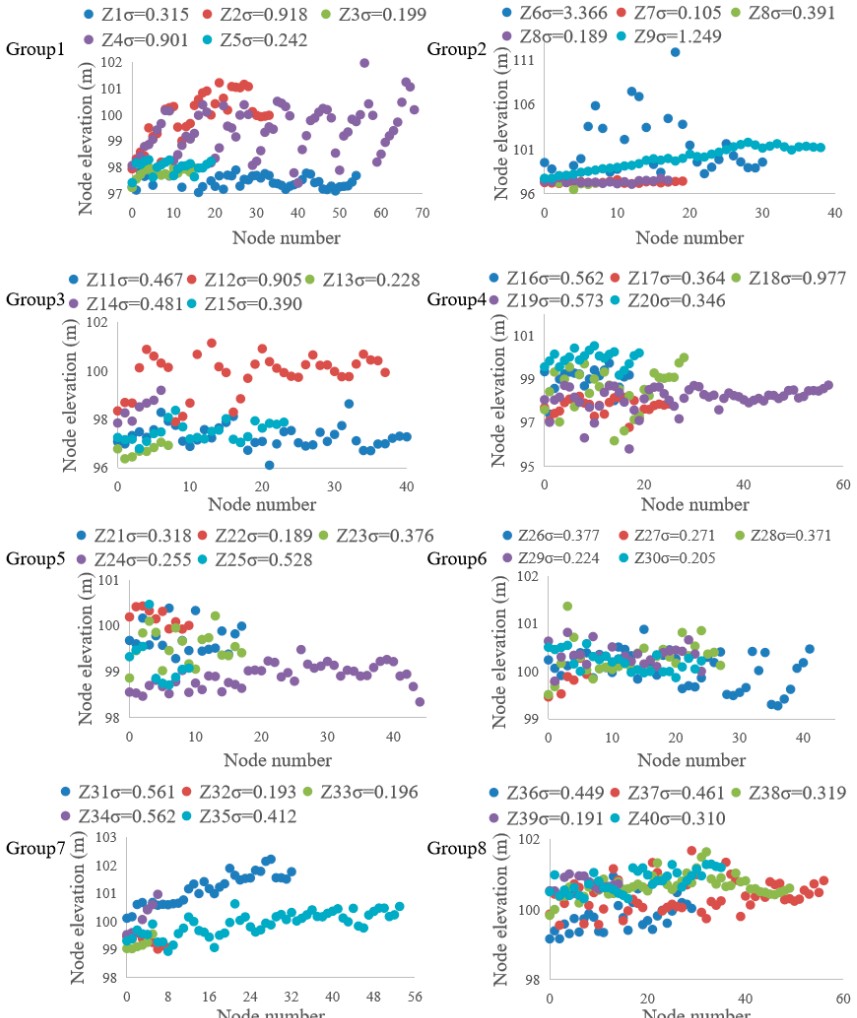

**Figure 8.** Elevation standard deviation of the original subcatchments.

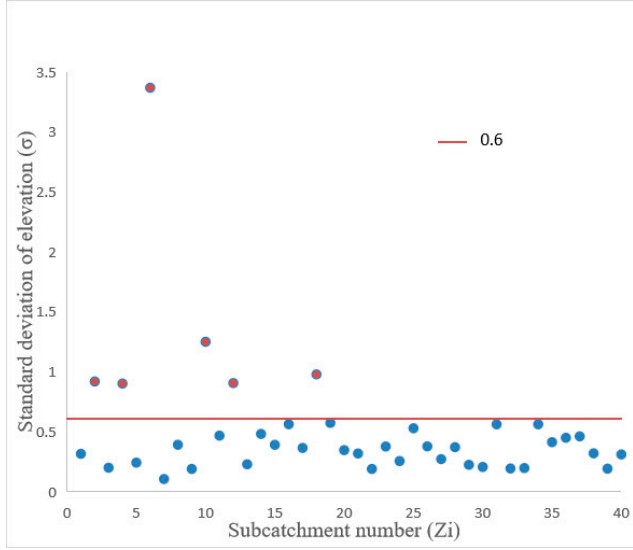

**Figure 9.** Subcatchments that need to be further divided (red dots) and subcatchments areas that needn't further division (blue dots).

The internal hydraulic exchange conditions of the six original subcatchments are determined based on the slope and aspect information (Figure 10), and are further divided by the method of identifying subcatchments by DEM, as shown in Figure 11.

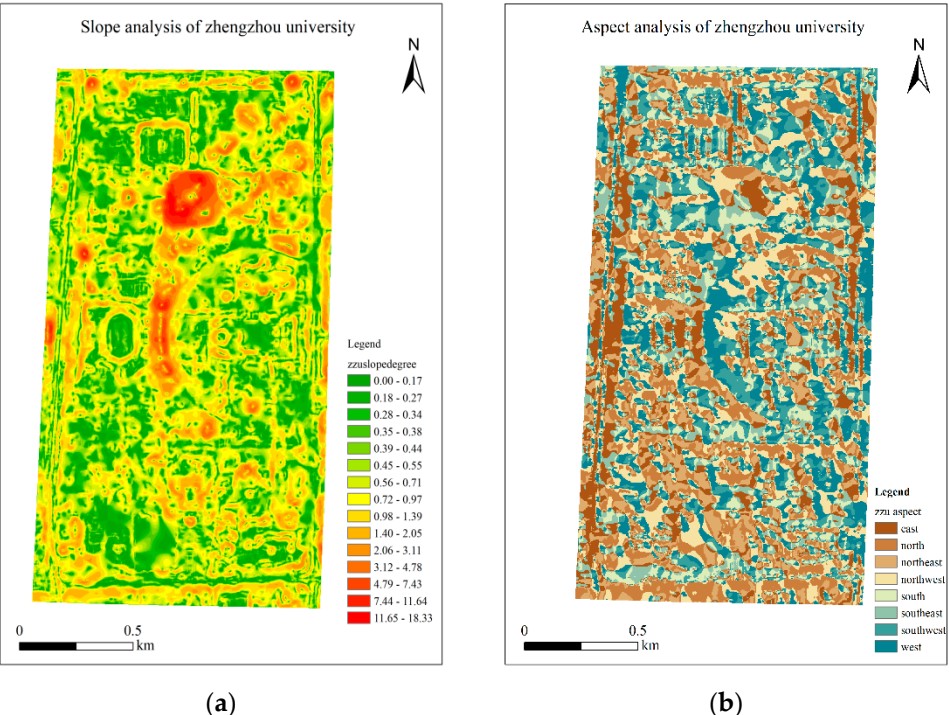

(**a**)    (**b**)

**Figure 10.** Slope map (**a**) and aspect map (**b**) of the study area.

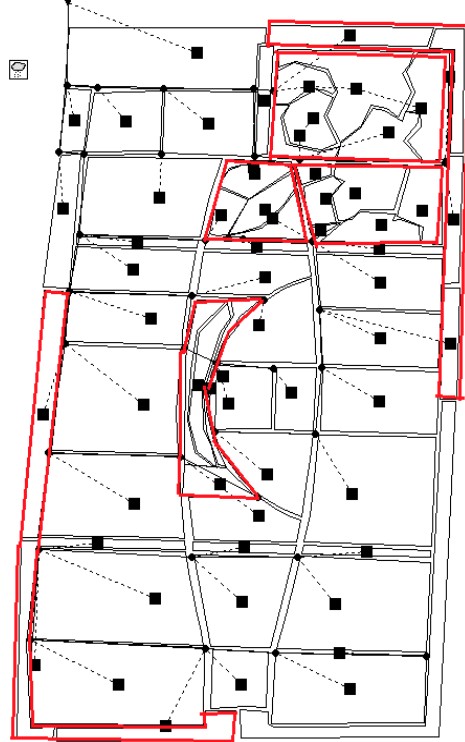

**Figure 11.** Division results of the new subcatchments.

The six areas enclosed by red lines in Figure 11 are the six original subcatchments that are not representative of the average slope. Based on field surveys, these six areas are further divided into 28 new subcatchments based on slope and aspect conditions. Ten of them flow to adjacent subcatchments, and the rest flow to adjacent nodes, and only one flowed to adjacent subcatchments before the improvement. The new subcatchments subdivided from the original subcatchments have different converging directions. This is a result that the original subcatchment area cannot perform, indicating that the new model not only conforms to the actual underlying surface in terms of parameters. The conditions also clarify to some extent the hydraulic exchange between the subcatchment and the nodes.

*4.2. Comparison of Model Simulation Results*

According to the two methods, the SWMM model of Zhengzhou University was constructed. Six rainfalls of 20140524, 20150707, 20160605, 20170812, 20180801 and 20180818 were selected for runoff simulation. Table 5 gives the relevant information of the six rainfalls. The six rainfalls include multiple recurrence periods from less than one year to nearly ten years, indicating that the data are reasonable and can comprehensively explain the problem.

**Table 5.** Rainfall information.

| Events | Time | Rainfall (mm) | Duration (min) | Recurrence Period (a) |
|---|---|---|---|---|
| 20140524 | 24 May 2014 | 31 | 330 | 0.45 |
| 20150707 | 7 July 2015 | 37 | 60 | 1.15 |
| 20160605 | 5 June 2016 | 85.5 | 460 | 6.21 |
| 20170812 | 12 August 2017 | 63.5 | 110 | 4.44 |
| 20180801 | 1 August 2018 | 63 | 60 | 9.67 |
| 20180818 | 18 August 2018 | 14 | 150 | 0.14 |

The water level process line of the northwest drain of the campus before and after the improvement is shown in Figure 12. In order to conveniently compare the simulation results of the two methods, the same parameter calibration results are used before and after the improvement. Among them, the red line is the original simulation result, and the blue line is the improved model simulation result. The blue line is located below the red line, which indicates that the improved runoff and flood peak flow have increased to a certain extent, and the peak time is relatively early. Compared with the actual runoff process (purple line), it was found that the improved model is closer to the actual runoff process, which shows that this method has a certain improvement significance to the "Natural-artificial boundary method". In comparison, the actual runoff process has more obvious "fast rise and fall" characteristics, and the improved model can better reflect this characteristic than before, which also shows that the original method slowed the terrain and flood process. Because not all the original subcatchments were subdivided; the improved model could not fully meet the actual runoff process. Subdividing more original subcatchments according to this method is beneficial to improve the accuracy of the simulation. In actual work, we can choose a suitable original subcatchment slope standard deviation threshold σ, and reduce it as much as possible without excessively increasing the workload. The small σ value can subdivide the original subcatchment area as much as possible, and reasonably divide the subcatchment area to improve the accuracy of the simulation.

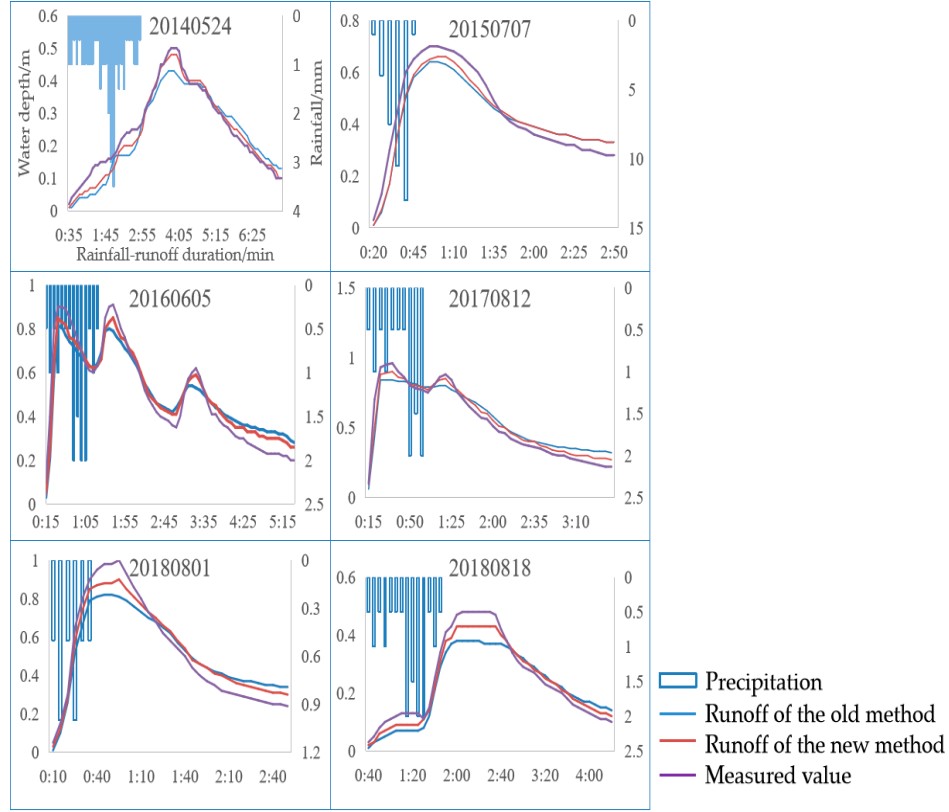

**Figure 12.** Comparison of simulation results of two methods for six rainfall events.

## 5. Discussion

By calculating the Nash-Sutcliffe efficiency coefficient [24] of the six runoff simulation results before and after the improvement, the rationality of the method improvement can be measured. The results are shown in Table 6. The Nash-Sutcliffe efficiency coefficient of all simulation results are greater than 0.85, indicating that both methods are suitable for the division of subcatchments, and the improved results are generally higher than before, and all results are above 0.9. The improved method performs better in the division of urban subcatchments and flood simulation, it is a reasonable improvement method.

**Table 6.** Nash-Sutcliffe efficiency coefficient of six-field runoff simulation.

| Events | Nash-Sutcliffe Efficiency Coefficient | | Increase/Decrease after Improvement |
|---|---|---|---|
| | **Before Improvement** | **After Improvement** | |
| 20140524 | 0.890 | 0.959 | ↑0.069 |
| 20150707 | 0.890 | 0.917 | ↑0.027 |
| 20160605 | 0.885 | 0.943 | ↑0.058 |
| 20170812 | 0.906 | 0.965 | ↑0.059 |
| 20180801 | 0.886 | 0.954 | ↑0.068 |
| 20180818 | 0.851 | 0.958 | ↑0.107 |

Due to the extensive and centralized distribution of infrastructure in urban areas, the flow path of natural runoff has changed greatly. Watersheds are the boundaries of natural drainage basin. Because the slope and aspect of the watershed are different on both sides, the runoff directions on the two sides are different. Due to the blocking and guiding effect of roads and pipe networks on water flow in urban areas, subcatchments divided by watersheds cannot fully reflect the actual flow path. Therefore, in the runoff simulation, roads and pipe networks are often used to divide subcatchments.

However, the result of such a division is that the slope and aspect of each subcatchment are averaged, but the direction of the flow is a vector and cannot be averaged, which makes this method problematic in the runoff, and there is often an error in calculating the flow of a subcatchment with large terrain fluctuations using the average slope.

For a long time, some scholars have studied the impact of increasing the spatial resolution of urban area model construction on flood simulation results, and have come to the conclusion that unilaterally improving the model's spatial resolution has little effect on flood simulation results [16,25]. This conclusion seems unexpected. Through this study we can give a reasonable explanation. The actual urban hydrological process is very complicated, and the hydrological phenomena reflected by different spatial scales are obviously different. In principle, the improvement of spatial resolution should be able to change the simulation results of urban floods. However, if it is not from very coarse resolution (kilometer-level) to very fine resolution (meter-level), it is difficult to find changes in hydrological phenomena from macro to micro. A large number of artificial structures and micro-topography in urban areas have greatly changed the natural runoff field, and their spatial scales are in the order of meters or even centimeters. Therefore, if the spatial resolution of the model is not fine to the meter level, it cannot represent urban hydrology. Additionally, the macroscopic features are difficult to be significantly improved by changing the spatial resolution. Therefore, the spatial resolution is increased from kilometers to 100 meters and ten meters, and it is essentially impossible to find a large change in the results of flood simulation.

To sum up, dividing the subcatchment area only by road-pipe network as the boundary cannot achieve high simulation accuracy. Although the model with high spatial resolution can improve the simulation accuracy, it requires spatial data below the meter level, which greatly improves the difficulty of obtaining data. This study considers the terrain fluctuation within the subcatchment with road-pipe network as the boundary. For subcatchments with large slope fluctuations and many changes in aspect, it is further subdivided into multiple subcatchments based on their slope characteristics. The new subcatchment area with uniform slope and small slope fluctuations not only reflects the actual runoff path to a certain extent, but also improves the accuracy of flow calculation. At the same time, it does not need to obtain high-precision underlying surface data, which reduces the workload.

## 6. Conclusions

This study explores the reasonable division of subcatchments for urban flood simulation. Currently three popular methods, "Natural-artificial boundary method", "Land use method" and "Tyson polygon method" all have their own scope, advantages and disadvantages. Their common feature is that the subcatchments are divided according to the two-dimensional ground surface and use the average slope to reflect the topographic characteristics of the subcatchment area. This treatment not only ignores the three-dimensional characteristics of the underlying surface, but also cannot describe the actual hydraulic exchange conditions between the subcatchments and the nodes, which brings errors to the model simulation.

In the case study of the new campus of Zhengzhou University, based on the DEM data obtained from actual sampling, the standard deviations of elevations of 40 original subcatchments divided by the "Natural-artificial boundary method" were analyzed, of which six obviously large, indicating that the average slope is not representative, and need to be further divided to improve the rationality of parameters and runoff paths. Based on the DEM data, the six original subcatchments were automatically identified in ArcGis, and 28 new subcatchments were obtained based on the correction of the slope and aspect, and a new model was constructed.

By comparing the simulation results of the two methods, it is shown that the slope method is more accurate than the "Natural-artificial boundary method" under the same calibration parameters, showing the improvement of the new method. Compared with the two simulation results, the actual runoff process reflects the characteristics of "rapid rise and fall". The improved model can better reflect

this characteristic than before, which also shows that the average slope will make the terrain flatten and also makes the flood process line flatten.

Because all the original subcatchments were not subdivided, the improved model could not fully meet the actual runoff process. Subdividing more original subcatchments according to this method is beneficial to improve the accuracy of the simulation. In actual work, you can choose a suitable original subcatchment slope standard deviation threshold σ, and reduce it as much as possible without excessively increasing the workload. Small σ value to subdivide more original subcatchments, and improve the accuracy of simulation.

**Author Contributions:** Z.W. provided the overall idea and data, B.M. was responsible for the experimental design, H.W. was responsible for the data analysis, and C.H. provided the instruments and methods. All authors have read and agreed to the published version of the manuscript.

**Funding:** The work was supported by National Natural Science Foundation of China, Key Program (Grant No. 51739009).

**Acknowledgments:** The Authors gratefully acknowledge Chengcai Zhang who contributed to some of the numerical computations.

**Conflicts of Interest:** The authors declare that they have no known competing financial interests or personal relationships that could have appeared to influence the work reported this paper.

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
