# Peer review of "Study on the Improved Method of Urban Subcatchments Division Based on Aspect and Slope- Taking SWMM Model as Example"

_hydrology, doi:10.3390/hydrology7020026_

Round 1

Reviewer 1 Report

m3/m2 - put the numbers with superscript

The term subcatchment appears written in the manuscript in two for

l1/l2, h1/h2 etc/- put the numbers with subscript

line 177 - what is with "4. Results and Discussion" there?

Figure 4 is at a poor resolution. Can it be replaced by one at a better resolution?

lines 239, 252, 269, 333, 353 - quotations in the text are placed in the wrong format

Figure 10 - the legend is placed inside a graph which makes it difficult to read. It must be placed outside the graphics. Also, the descriptions for the XOY axes are either put in all the graphs or they are put in the legend or they are put in the first graph not the second one.

Author Response

Response to Reviewer 1 Comments

Point 1: m3/m2 - put the numbers with superscript

Response 1: Thank you for your review. I have made changes in lines 103-106 of the manuscript, as shown below.

        Where, V is the surface water collection volume, m3; d is the water depth, m; t is the time, s; A is the surface area, m2; i* is the net rainfall intensity, mm/s; Q is the outflow, m3/s; W is the sub-basin flood width, m; n is the surface Manning coefficient; dp is the maximum surface depression depth, m; S is the average slope of the subcatchment.

Point 2: The term subcatchment appears written in the manuscript in two for l1/l2, h1/h2 etc/- put the numbers with subscript

Response 2: I have made changes in lines 124-129 of the manuscript, as shown below.

        Where, u is the water flow velocity (m/s) at the boundary of two adjacent cells Zi, j and Zi, j + 1 with elevations l1 and l2 and side length r, g is the acceleration of gravity (9.8 m/s2), CD is the drag coefficient, h1 and h2 are the water depths of the two cells along the water flow direction. When the cell is small enough, according to the continuity characteristics of the water flow, it can be considered that the adjacent two cells have the same water depth, that is, h1 and h2 are equal to h, as shown in the figure 2, and the slope S is given by equation (4):

Point 3: line 177 - what is with "4. Results and Discussion" there?

Response 3: The "unit" here is redundant, and I have deleted it in the manuscript.

Point 4: Figure 4 is at a poor resolution. Can it be replaced by one at a better resolution?

Response 4: I have replaced it with a clearer one in the manuscript.

Point 5: lines 239, 252, 269, 333, 353 - quotations in the text are placed in the wrong format

Response 5: I have corrected it in the manuscript.

Point 6:  Figure 10 - the legend is placed inside a graph which makes it difficult to read. It must be placed outside the graphics. Also, the descriptions for the XOY axes are either put in all the graphs or they are put in the legend or they are put in the first graph not the second one.

Response 6: I have modified it as required, placing the legend outside and the descriptions for the XOY axes are put in the first graph.

Reviewer 2 Report

1)  The methodology presented here is complex as it links a number of steps to generate the outputs of the model. It would be helpful if the authors developed a logic (flow) chart that describes the linkages between the sequence of analytical steps in the methodology.

2) The text in the existing figured is a bit small could the authors enlarge the text to some degree or to the size required by the instructions.

3) I have done some editing which I hope is helpful with the folw of the text.

Author Response

Response to Reviewer 2 Comments

Point 1: The methodology presented here is complex as it links a number of steps to generate the outputs of the model. It would be helpful if the authors developed a logic (flow) chart that describes the linkages between the sequence of analytical steps in the methodology.

Response 1: Thank you for your review. I have added a flowchart in the manuscript to introduce the operation process, as shown on page 7.

Point 2: The text in the existing figured is a bit small could the authors enlarge the text to some degree or to the size required by the instructions.

Response 2: I have enlarged some of the figures so that the text can be clearly distinguished.

Point 3:I have done some editing which I hope is helpful with the folw of the text.

Response 3: Thank you for editing my manuscript, it is very helpful to me.

Reviewer 3 Report

The authors claim that they have improved subcatchment division for SWMM modeling. However, it is not clear how they actually improved the approach in Section 2.2 in the manuscript. Without this information, it is not easy to review the text. I strongly advise to rewrite Section 2.2 to clearly convey their improvement with a flowchart or pseudo code.

Please add a space after periods in the Abstract.

L24: "an model" => "a model"

L66: "However" => "however"

Eq. (2): Could you please explain why you need "- dp" in this equation and depict what "dp" means in a Figure?

L109: The error depends on the variability of the slope, not on the quantity of the slope. A uniformly steep subcatchment will have a higher average slope with a small deviation, which leads to a smaller error.

Eq. (4): Why did you use the slope length in the denominator instead of the horizontal distance "r" when you used "r" in the other term (h2-h1)/r? I know [20] used the same equations, however they didn't explain why either. What is the x coordinate? Along the bed slope or the horizontal distance? It seems to me that an assumption is missing where for a small slope, l1, l2, h1, h2 are approximately the same in the vertical direction and in the direction perpendicular to the channel slope. However, again, sqrt((l1-l2)^2+r^2) is along the channel bottom, but r is the horizontal distance in Eq. (3). It may be negligible, but it needs to be clearly mentioned and explained why.

L157: "exist huge difference" => "exhibit a huge difference"?

L161: How are the "natural-artificial boundary method" and the "DEM identification sub-basin method" different? Please clarify it in the text. Maybe, using Figure 5?

L166-169: "when [...], and the slope" => "when [...], the slope"? Or did you mean "when [...], and the slope [...]. The slope is not representative." => "when [...], and the slope [...], the slope is not representative"? Based on the context, the former? Please rewrite this sentence.

After reading up to this point, it is still not clear what your proposed method is. Section 2.2 does not make it clear how your improved method works. Just mentioning the combination of the natural-artificial boundary method and DEM identification sub-basin model is not enough. I cannot further review the manuscript without this critical information.

Author Response

Response to Reviewer 3 Comments

Point 1:The authors claim that they have improved subcatchment division for SWMM modeling. However, it is not clear how they actually improved the approach in Section 2.2 in the manuscript. Without this information, it is not easy to review the text. I strongly advise to rewrite Section 2.2 to clearly convey their improvement with a flowchart or pseudo code.

Response 1: Thank you for your review. I have rewritten chapter 2.2 and added a flowchart to more clearly represent the process of subcatchment division.

Point 2:Please add a space after periods in the Abstract.

Response 2: I have made corresponding changes in the Abstract.

Point 3: L24: "an model" => "a model"  L66: "However" => "however"

Response 3: Thank you for pointing out these two grammatical errors, I have made changes in the manuscript.

Point 4: Eq. (2): Could you please explain why you need "- dp" in this equation and depict what "dp" means in a Figure?

Response 4: Where dpis the maximum depth of surface water storage, when d> dp, there will be slope confluence, and the water within the depth of dp will not be discharged into the pipeline or river.  I have modified and added this part in the manuscript

Point 5: L109: The error depends on the variability of the slope, not on the quantity of the slope. A uniformly steep subcatchment will have a higher average slope with a small deviation, which leads to a smaller error.

Response 5: There is no doubt that your view is correct. For urban areas, the aspect of the subcatchments surrounded by roads and rivers often changes, so uniformly steep subcatchment is very rare. According to Fig. 8 (b), it can be seen that the slope aspect in the study area changes rapidly, and there is no uniform and steep subcatchment. Therefore, the elevation standard deviation of the subcatchment can be used to represent the degree of topography relief in the sub-catchment.

Point 6: Eq. (4): Why did you use the slope length in the denominator instead of the horizontal distance "r" when you used "r" in the other term (h2-h1)/r? I know [20] used the same equations, however they didn't explain why either. What is the x coordinate? Along the bed slope or the horizontal distance? It seems to me that an assumption is missing where for a small slope, l1, l2, h1, h2 are approximately the same in the vertical direction and in the direction perpendicular to the channel slope. However, again, sqrt((l1-l2)^2+r^2) is along the channel bottom, but r is the horizontal distance in Eq. (3). It may be negligible, but it needs to be clearly mentioned and explained why.

Response 6: When I quoted [20], I think that the formula of slope S is wrong. After your doubts, I have strengthened this suspicion, because from the definition of slope, the denominator of Eq. (3)and (4) should be the horizontal distance r, not sqrt((l1-l2)^2+r^2). Therefore, I modified the Eq. (3)and (4) and used the assumption you mentioned as the premise of Eq. (5).

Point 7: L157: "exist huge difference" => "exhibit a huge difference"?

Response 7: Yes, this is a syntax error, I have corrected it.

Point 8: L161: How are the "natural-artificial boundary method" and the "DEM identification sub-basin method" different? Please clarify it in the text. Maybe, using Figure 5?

Response 8: Yes, it is found from Figure (5) thatthe “Natural-artificial boundary method“ is based on the road and water system as the boundary, which can reflect the distribution of the road-water system of the underlying surface. The second method is to identify the watershed based on DEM data and divide the underlying surface into subwatersheds. A detailed explanation of this issue is on lines 270-281.

Point 9: L166-169: "when [...], and the slope" => "when [...], the slope"? Or did you mean "when [...], and the slope [...]. The slope is not representative." => "when [...], and the slope [...], the slope is not representative"? Based on the context, the former? Please rewrite this sentence.

Response 9: Yes, this sentence is easy to cause ambiguity, I have rewritten in the manuscript line 173-181, as shown below.

        Due to the change of the slope direction in the subcatchment area, when the standard deviation of the elevation of the subcatchment area is not greater than 0.6, the terrain is considered to be gentle, and the average slope can be used as the slope parameter. Otherwise, the terrain changes significantly, and it is necessary to further divide the original subcatchment according to the method of DEM automatic subcatchment identification, so as to obtain new subcatchments with more representative slope parameters. It should be noted that the subcatchment elevation standard deviation threshold σ is a value defined according to modeling needs. The smaller the value, the finer the subcatchments division, and the higher the accuracy of the simulation, but model building will be more difficult.

Point 10: After reading up to this point, it is still not clear what your proposed method is. Section 2.2 does not make it clear how your improved method works. Just mentioning the combination of the natural-artificial boundary method and DEM identification sub-basin model is not enough. I cannot further review the manuscript without this critical information.

Response 9: I have made corresponding amendments in the manuscript. Thank you again for your review and valuable comments. This has played a very good role in improving my manuscript.

Round 2

Reviewer 3 Report

L120: "Eq. is" => which equation?

L134: "Eq. indicates" => which equation?

Figure 6 needs to be fixed and placed properly.

Figure 4. Some text is outside the diagrams.

Figure 8: I cannot understand this figure. What does it show? The elevation standard deviation of the original subcatchments? (Figure 7a?) It looks like there are more than 40 points in different colors. Why are there more than one of each Z in these plots? What about node numbers? I thought Zs are node numbers, but these Zs are assigned to multiple nodes? What do you mean by groups here? You never mentioned Groups in the text. This figure is very confusing. Please clarify what all these terms mean in the text.

Figure 9 seems to make sense.

L284-286: Where is the main clause? "In order to make ..., improve, ... calculate" what did you?

Author Response

Point 1: L120: "Eq. is" => which equation?

Response 1: Thank you for your review. "Eq. is" => "Eq. (3)",I have made a correction in the manuscript.

Point 2: L134: "Eq. indicates" => which equation?

Response 2: "Eq. indicates" => "Eq. (5)",I have made a correction in the manuscript.

Point 3: Figure 6 needs to be fixed and placed properly.

Response 3: Yes, it looks like this figure has been divided into two parts, and I have corrected it in the manuscript.

Point 4: Figure 4. Some text is outside the diagrams.

Response 4: I revised this figure so that the text is within the border as much as possible to facilitate reading.

Point 5: Figure 8: I cannot understand this figure. What does it show? The elevation standard deviation of the original subcatchments? (Figure 7a?) It looks like there are more than 40 points in different colors. Why are there more than one of each Z in these plots? What about node numbers? I thought Zs are node numbers, but these Zs are assigned to multiple nodes? What do you mean by groups here? You never mentioned Groups in the text. This figure is very confusing. Please clarify what all these terms mean in the text.

Response 5: Thank you for your comments. The scattered points in this figure represent the elevation of sampling points every 2m in the study area. These sampling points are represented by different colors in different original subcatchments. Zi is the number of the original subcatchment, indicating the i-th sub,the 40 original subcatchments are numbered Z1, Z2, Z3, ..., Z40. For the convenience of display, I have divided them into 8 groups of 5 each. The legend indicates the elevation standard deviation of all sampling points in each subcatchment. I have explained this part in the manuscript.

Point 6: L284-286: Where is the main clause? "In order to make ..., improve, ... calculate" what did you?

Response 6: This sentence is inappropriate in this part. This part mainly talks about how I get the process of the original subcatchments that needs to be further divided, not how to improve the accuracy of the model. Therefore, I deleted this sentence.
